# Hypoglycemic and Antioxidant Properties of Extracts and Fractions from Polygoni Avicularis Herba

**DOI:** 10.3390/molecules27113381

**Published:** 2022-05-24

**Authors:** Kun Zhang, Mei Han, Xia Zhao, Xuelin Chen, Hanlei Wang, Jiyan Ni, Yumei Zhang

**Affiliations:** 1Key Laboratory of Tropical Plant Resource and Sustainable Use, Xishuangbanna Tropical Botanical Garden, Chinese Academy of Sciences, Kunming 650223, China; zhangkun@xtbg.ac.cn (K.Z.); hanmei@xtbg.ac.cn (M.H.); zhaoxia@xtbg.ac.cn (X.Z.); chenxuelin@xtbg.ac.cn (X.C.); wanghanlei@xtbg.ac.cn (H.W.); nijiyan@xtbg.ac.cn (J.N.); 2University of Chinese Academy of Sciences, Beijing 100049, China

**Keywords:** Polygoni Avicularis Herba, hypoglycemic, antioxidant, flavonoids, phenolics, molecular docking

## Abstract

Our research focused on the hypoglycemic capability and the possible mechanisms of extract and fractions from Polygoni Avicularis Herba (PAH) based on *α*-glucosidase, *α*-amylase inhibition assays, glucose uptake experiment, HPLC-MS analysis, and molecular docking experiment. In addition, DPPH, ABTS, and FRAP assays were used for determining the antioxidant capability. The results of total flavonoids and phenolics contents showed that ethyl acetate fraction (EAF) possessed the highest flavonoids and phenolics with values of 159.7 ± 2.5 mg rutin equivalents/g and 107.6 ± 2.0 mg galic acid equivalents/g, respectively. The results of in vitro hypoglycemic activity showed that all samples had effective *α*-glucosidase inhibition capacities, and EAF possessed the best inhibitory effect with IC_50_ value of 1.58 ± 0.24 μg/mL. In addition, n-butanol fraction (NBF) significantly promoted the glucose uptake rate of 3T3-L1 adipocytes. HPLC-MS analysis and molecular docking results proved the interactions between candidates and *α*-glucosidase. The results of antioxidation capacities showed that EAF possessed the best antioxidation abilities with DPPH, ABTS, and FRAP. In summary, the hypoglycemic activity of PAH might be related to the inhibition of *α*-glucosidase (EAF > PEF > NBF) and the promotion of glucose uptake in 3T3-L1 adipocytes (NBF). Simultaneously, the antioxidation capacity of PAH might be related to the abundant contents of flavonoids and other phenolics (EAF > PEF > NBF).

## 1. Introduction

Polygoni Avicularis Herba (PAH) is the dry aerial part of *Polygonum aviculare* L. (family Polygonum). It is used in traditional Chinese medicine to treat dysuria, abdominal pain caused by intestinal parasites, skin eczema, and genital itching [1]. According to the literature, PAH possessed anti-inflammatory [2,3], antibacterial [3,4], antioxidant [5,6,7], anti-obesity [8,9], hypoglycemic [10,11], and vasorelaxant [12] activities. The main bioactive components obtained in PAH were flavonoids, phenolic acids, alkaloids, terpenes, sterols, quinones [13]. Flavonoids were the main components obtained from PAH, including quercetin, myricetin, kaempferol, olivine glycosides, myricetin 3-*O*-(3″-*O*-galloyl)-rhamnopyranoside, kaempferin, rutin, hyperoside, juglansin, and luteolin [14].

Diabetes mellitus (DM), a metabolic disorder characterized by hyperglycemia induced by insulin secretion deficiency and/or resistance to its action, affects millions of people around the world [15]. At present, antidiabetic drugs such as biguanides, sulfonylureas, meglitinides, thiazolidinediones, dipeptidyl peptidase IV inhibitors, and *α*-glucosidase inhibitors have many side effects, such as weight gain, hypoglycemia, gastrointestinal disorders, liver and kidney damage, and hypersensitivity reactions [15]. On the other hand, plants rich in certain types of flavonoids and other phenolics can exert a significant impact on diabetes via protecting pancreatic islet B cells, reducing the absorption of glucose in the digestive tract, and promoting glucose uptake in adipocytes [16,17,18]. Therefore, natural products from medicinal plants might have a good development prospect in the field of diabetes treatment.

Antioxidant activity has been proved to be relevant in the treatment of Alzheimer’s disease [19], diabetes [20], hypertension [21], lung fibrosis [22], and tumors [23]. Therefore, natural products possessing antioxidant activity may present a complementary alternative for the treatment of these diseases. The previous literature reported that the total phenolics and flavonoids content of PAH ethanolic extract (PAHEE) was 677.4 ± 62.7 mg/g and 112.7 ± 13.0 mg/g, respectively [5], and the antioxidant effects of the extract were proved by free radical scavenging assay, superoxide radical scavenging assay, lipid peroxidation assay, and hydroxyl radical-induced DNA strand scission assay [5,7]. However, there are few reports on the total flavonoids and phenolics contents of PAHEE fractions and their corresponding antioxidant activity determined by FRAP and ABTS assays.

Previous evidence showed that PAH had effective antioxidant and hypoglycemic activities [6,7,10,11]. Nevertheless, there are few comparative studies on the antioxidant, hypoglycemic activities, and the possible hypoglycemic mechanisms and molecules of its fractions. Therefore, in this study, the antioxidant, hypoglycemic properties, and the possible hypoglycemic mechanisms of the active fractions from PAHEE are reported. At first, we detected the total flavonoids and total phenolics content in different fractions, and the fractions were then assayed for their antioxidant potential via 2,2-diphenyl-2-picrylhydrazyl (DPPH), ABTS, and FRAP assays. Next, in vitro hypoglycemic assays (*α*-glucosidase, and *α*-amylase inhibitory activities, as well as glucose uptake in 3T3-L1 adipocytes experiment), along with HPLC-MS and in silico studies were utilized to explore the antidiabetic mechanisms of PAH.

## 2. Results

### 2.1. Total Flavonoids Content

The standard curve of rutin was y = 0.0054x + 0.047, *R*^2^ = 0.9994 (Appendix A). The quantitative analysis results of total flavonoids of PAHEE and its fractions showed that total flavonoids content in different fractions had significant differences (*p* < 0.05). Ethyl acetate fraction (EAF) possessed the highest content of flavonoids with values of 159.7 ± 2.5 mg/g, followed by petroleum ether fraction (PEF) and PAHEE (88.5 ± 4.4 and 68.7 ± 0.3 mg/g, respectively). n-Butanol fraction (NBF) exhibited the lowest total flavonoids content (Table 1).

### 2.2. Total Phenolics Content

The standard curve of gallic acid was y = 0.0463x + 0.0173, *R*^2^ = 0.9995 (Appendix A). Consistent with the tested results of total flavonoids content, EAF possessed the highest content of phenolics with values of 107.55 ± 1.96 mg/g, followed by PEF and PAHEE (62.2 ± 1.1 and 43.2 ± 0.6 mg/g, respectively). NBF had the lowest total phenolic content (Table 1).

### 2.3. The Inhibitory Effects of PAHEE and Its Fractions on α-Glucosidase and α-Amylase

*α*-Glucosidase inhibition abilities of PAHEE and its’ fractions were assayed, acarbose was used as a positive control. In a dose-dependent manner, the PAHEE, PEF, EAF, and NBF (Figure 1) exhibited effective *α*-glucosidase inhibition activities, with IC_50_ values from 1.58 to 4.25 μg/mL. EAF possessed potent *α*-glucosidase inhibition ability with IC_50_ value of 1.58 ± 0.24 μg/mL (Figure 1c), PEF and NBF showed comparable inhibition activities with IC_50_ values of 2.32 ± 0.10 and 2.57 ± 0.28 μg/mL, respectively (Figure 1b,d). The results of *α*-amylase inhibition experiment showed that NBF possessed a weak inhibitory effect on *α*-amylase with IC_50_ value of 4.73 ± 1.41 μg/mL, the other samples had no obvious inhibitory effect on *α*-amylase.

### 2.4. Enzyme Kinetic Equation

In order to evaluate the type of the fractions partitioned from PAHEE on *α*-glucosidase (the digestive enzyme that was best inhibited), Lineweaver–Burk plotting was performed. As shown in Figure 2, all data lines of PAHEE, PEE, EAE, and NBE on the Lineweaver–Burk plot intersected in a point in the third quadrant, and with the increase in inhibitor concentration, the kinetic parameters Vmax (longitudinal intercept is 1/Vmax) and Michaelis constant Km (cross-sectional distance −1/Km) decreased. Therefore, all inhibitory effects of samples on *α*-glucosidase enzyme belonged to the reverse-competitive inhibition type [24], which suggested that the inhibitors presented in samples PEE, EAE, and NBE might be bound to the enzyme–substrate complex to inhibit *α*-glucosidase.

### 2.5. Glucose Uptake and Cell Viability Assays

Fully differentiated 3T3-L1 adipocytes were used to detect the glucose uptake rates and cell viabilities of different groups (Model group, positive control group, PAHEE group, PEF group, EAF group, and NBF group), and insulin was considered as a positive control. Different from the antioxidant results, NBF showed a concentration-dependent effect of promoting glucose uptake in 3T3-L1 adipocytes (Figure 3b). The results showed that the glucose concentration in medium of each group was 5.5 mmol/L at 0 h, after 24 h of administration, the glucose concentration in medium of model group, 40, 80, and 160 μg/mL of NBF groups were 3.28 ± 0.19, 2.51 ± 0.01, 2.09 ± 0.05, and 1.82 ± 0.03 mmol/L, and the glucose uptake rates were (40.32 ± 5.87)%, (54.30 ± 0.49)%, (62.08 ± 0.87)%, and (66.93 ± 0.56)%, respectively. Concentrations of 40, 80, and 160 μg/mL of NBF had significant effects on promoting glucose uptake in 3T3-L1 adipocytes; the differences were statistically significant (* *p* < 0.01, *** *p* < 0.001, and *** *p* < 0.001, respectively). The results of cell viability showed that the insulin group and 160 μg/mL EAF group had weak inhibitory effects on the viability of 3T3-L1 adipocytes (*** *p* < 0.001). In addition, 80 μg/mL EAF group and 160 μg/mL NBF group had weak promotion effects on the viability of 3T3-L1 adipocytes (*** *p* < 0.001). Except for the groups mentioned above, other groups had no significant effects on cell viability (Figure 3c). In conclusion, NBF had a significant capability to promote glucose uptake in 3T3-L1 adipocytes, and without inhibitory effect on cell viability.

### 2.6. HPLC-MS Analysis of EAF

The primary constituents of PAH were reported to be flavonoids, phenolic acids, alkaloids, terpenes, sterols, quinones, other phenylpropanoids, etc. [13]. Consistent with the previous literature, the main components we detected from EAF were flavonoids, phenolics, alkaloids, quinones, and terpenes (Table 2). Flavonoids included agathisflavone, delphinidin 3-*O*-*β*-D-galactopyranoside, 3,5,7,2′,6′-pentahydroxyflavonol, 3′-methoxydaidzein, avicularin, cyanidin, delphinidin, pelargonidin, and 6-hydroxy kaempferol-7-*O*-glucoside. Phenolics included 2-methoxy-4-(3-methoxy-1-propenyl)-phenol (Table 3). In these compounds, 3,5,7,2′,6′-pentahydroxyflavonol, agathisflavone, avicularin, and delphinidin were reported to exhibited antioxidant activity; agathisflavone, avicularin, delphinidin, cyanidin, pelargonidin, delphinidin-3-arabinoside, pelargonidin-3-galactoside, and leonurine were reported to exhibited hypoglycemic activity; and the *α*-glucosidase inhibitory effects of the other flavonoids and phenolics was not reported, as shown in Table 2.

### 2.7. Molecular Docking of the Candidate Compounds on α-Glucosidase

The molecular docking results showed that flavonoids agathisflavone and delphinidin 3-O-β-D-galactoside possessed the lowest binding energy with α-glucosidase (−11.35, −11.58 kcal/mol, respectively). In addition, the other flavonoids and phenolics also showed inhibitory effect on α-glucosidase with superior binding energy (Table 2). The surface structure of ligand–enzyme complexes showed that the candidate was positioned in the pocket of α-glucosidase, as illustrated in Figure 4a,c. 

The docking results showed that agathisflavone formed two hydrogen-bonding interactions between GLU231, LYS225 residues of *α*-glucosidase and hydroxyl groups, delphinidin 3-*O*-*β*-D-galactoside formed two hydrogen-bonding interactions between PHE516, ALA518 residues of the enzyme and hydroxyl groups, and the inhibitory effects of delphinidin 3-*O*-*β*-D-galactoside, 3,5,7,2′,6′-pentahydroxyflavone, 5,7,2′,3′-tetrahydroxyflavone, 3′-methoxydaidzein, 6-hydroxykaempferol-7-*O*-glucoside, delphinidin 3′-*O*-(2″-*O*-galloyl-*β*-galactoside), and peonidin on α-glucosidase via molecular docking were reported for the first time (Table 2). In summary, the results indicated that the flavonoids and phenolics detected in EAF might be bound to the active site of *α*-glucosidase to inhibit the activity of the enzyme.

### 2.8. DPPH-Free Radical Scavenging Assay

DPPH-free radical scavenging assay is widely used to determine the antioxidant capacity of natural products [37]. PAHEE and its fractions possessed effective scavenging capacities on DPPH-free radicals, and had concentration-dependent relationships. Ascorbic acid was considered as a positive control with the SC_50_ value of 2.4439 ± 0.33 μg/mL (Figure 5e). EAF had the strongest scavenging capacity with the SC_50_ value of 19.94 ± 1.37 μg/mL (Figure 5c). Simultaneously, the scavenging effect of PEF (SC_50_ = 31.13 ± 2.23 μg/mL) was stronger than PAHEE (SC_50_ = 47.64 ± 3.72 μg/mL). In summary, the results showed that the DPPH free radical scavenging ability was EAF > PEF > PAHEE > NBF, and the scavenging capacity of PAH was related to the accumulative effects of each fraction.

### 2.9. ABTS Radical Scavenging Assay

The ABTS radical scavenging assay is commonly used to detect the scavenging effect of the sample on ABTS radical [38,39]. All of the samples showed ABTS radical scavenging capacity in vitro, ranging from 0.42 to 1.30 mmol Trolox/L, as shown in Table 3 EAF possessed the strongest ABTS radical scavenging ability (1.30 ± 0.003 mmol Trolox/L), while NBF and PAHEE showed a lower antioxidant potential than PEF (1.12 ± 0.012 mmol Trolox/L).

### 2.10. Ferric Reducing Antioxidation Power

Antioxidants presented in samples have the ability to reduce ferric tripyridyltriazine (Fe ^3+^ TPTZ) into ferrous tripyridyltriazine (Fe ^2+^ TPTZ) to evaluate the antioxidant potential [40]. The tested results were consistent with the ABTS assay, all of the samples showed ferric reducing antioxidation power (Table 3), EAF possessed the strongest ferric reducing antioxidation power with 0.76 ± 0.036 mmol Trolox/L, while NBF and PAHEE showed lower ferric reducing antioxidation power than PEF (0.54 ± 0.004 mmol Trolox/L).

## 3. Discussion

The hypoglycemic activity of PAH has been reported in traditional Chinese medicine classics [41]. In subsequent studies, Zhao, et al. and Chen, et al. confirmed the hypoglycemic effect of PAH in clinical trials, but did not clarify its possible hypoglycemic mechanisms [10,11]. Up to now, there have been few reports on the hypoglycemic mechanism of PAH. In our research, we explored the hypoglycemic mechanism of PAH via *α*-glucosidase inhibition assay, *α*-amylase inhibition assay, and 3T3-L1 adipocytes glucose uptake experiments. The results showed that the polar fractions of PAHEE might have different contributions for diabetes treatment. The digestive enzymes (*α*-glucosidase, *α*-amylase) inhibition experiments showed that all samples (PAHEE, PEF, EAF, and NBF) have effective *α*-glucosidase inhibition activity, and NBF possessed a weak *α*-amylase inhibition activity.

3T3-L1 cell line is considered a classic cell line that has been frequently used in research such as adipocytes differentiation, glucose uptake, and lipid metabolism [42]. In recent years, more and more studies have reported the hypoglycemic activity of the plant extracts or active ingredients via glucose uptake experiments in 3T3-L1 adipocytes [43,44]. Our results showed that NBF possessed a significant dose-dependent effect on promoting glucose uptake in 3T3-L1 adipocytes, and the difference was statistically significant (*** *p* < 0.001). Interestingly, NBF showed the weakest antioxidant and *α*-glucosidase inhibition activities, but showed the strongest effect of promoting glucose uptake in 3T3-L1 adipocytes. This result was discovered for the first time in PAH, as also its active ingredients and the mechanisms of promoting glucose uptake. Nevertheless, further studies should be needed to complete these data.

The antioxidant activity of PAH crude extracts and fractions have been reported in the literature by different methods, including DPPH free radical scavenging assay, superoxide scavenging assay [5,45], hydroxyl radical scavenging assay, and total reducing ability tests [7]. Our study evaluated PAHEE and its polar fractions’ antioxidant abilities via DPPH, ABTS, and FRAP assays. The results of DPPH free radical scavenging were consistent with the previous literature, EAF > PEF > PAHEE > NBF, ABTS, and FRAP results proved this point as well. It is noteworthy that the conclusion was consistent with total flavonoids and total phenolic content results, and higher content of flavonoids and phenolics showed better antioxidant activity, which indicated that the antioxidant capacity of PAH might be related to its abundant flavonoids and phenolics.

Polyphenols such as flavonoids and tannins can inhibit the digestion of carbohydrates to glucose by inhibiting the activity of key enzymes, such as α-glucosidase and α-amylase [46]. According to the literature, some of the flavonoids detected by HPLC-MS in our study have been reported to have potential hypoglycemic and antioxidant activity. On the one hand, the results showed that agathisflavone, avicularin, cyanidin, and pelargonidin exhibited significant inhibitory activity in digestive enzymes (*α*-glucosidase and/or *α*-amylase), and agathisflavone, avicularin inhibited *α*-glucosidase with IC_50_ values of 11.4 ± 0.9 µmol/L, 69.8 mg/L, respectively. In addition, delphinidin-3-arabinoside has the potential to modulate dipeptidyl peptidase-IV and its substrate GLP-1, to increase insulin secretion [26,27,28,29,30,31,32,33,34]. On the other hand, previous research suggested that 3,5,7,2′,6′-pentahydroxyflavone with a hydroxyl group in the B ring possessed a potent activity against lipid peroxidation [25]. In summary, the results suggested that PAH might possess great hypoglycemic and antioxidant activities, and its hypoglycemic activity might be related to the inhibition of digestive enzymes, and the promotion of insulin secretion.

Our results proved that flavonoids and phenolics were the abundant ingredients presented in EAF (the strongest α-glucosidase inhibitory effect). Next, we virtual-docked the detected compounds (aglycones and glycosides of flavonoids and other phenolics) with the crystal structure of *α*-glucosidase via AutoDock in silico. The results showed that the candidate compounds might be bound to the active site of *α*-glucosidase to inhibit the activity of *α*-glucosidase. It is worth noting that the α-glucosidase inhibitory abilities of delphinidin 3-*O*-*β*-D-galactoside, 3,5,7,2′,6′-pentahydroxyflavone, 5,7,2′,3′-tetrahydroxyflavone, 3′-methoxydaidzein, and delphinidin 3′-*O*-(2″-*O*-galloyl-*β*-galactoside) peonidin via molecular docking were reported for the first time, as shown in Table 3. Furthermore, several polyphenols, such as resveratrol, epigallocatechin-3-gallate and quercetin, enhanced glucose uptake in the muscles and adipocytes by translocating GLUT4 to plasma membrane mainly by the activation of the AMP-activated protein kinase pathway [46]. In summary, it could be seen that the in vitro hypoglycemic activity of PAH might be related to the inhibition of *α*-glucosidase (EAF, PEF, and NBF) and the promotion of glucose uptake in 3T3-L1 adipocytes (NBF).

## 4. Materials and Methods

### 4.1. Chemicals and Reagents

3T3-L1 mouse preadipocytes were purchased from the American Type Culture Collection (ATCC, Manassas, VA, USA). High glucose DMEM, low glucose DMEM, Pen-Strep solution (P/S), insulin, certified fetal bovine serum (FBS), special newborn calf serum (NBCS), and phosphate buffered saline (PBS) were purchased from Biological Industries (Shanghai, China). The glucose test kit was purchased from Rongsheng Biotech Co., Ltd. (Shanghai, China). *α*-Glucosidase (solid), 3,5-dinitrosalicylic acid, *p*-nitrophenyl *α*-D-glucopyranoside (PNPG), and ascorbic acid were purchased from Yuanye Biotech Co., Ltd. (Shanghai, China). Acarbose and rutin were obtained from Solarbio (Beijing, China). CellTiter 96^®^ AQueous One Solution Reagent (Promega Corporation, Madison, WI, USA). DPPH, ABTS, and FRAP detection reagents were purchased from Suzhou Comin Biotechnology Co., Ltd. (Jiangsu, China). Sodium nitrite, aluminum nitrate, sodium carbonate, and sodium hydroxide were purchased from MACKLIN (Shanghai, China).

### 4.2. Preparation of Plant Extracts

Polygoni Avicularis Herba (*Polygonum aviculare* L.) was purchased from Hele Chinese Medicine Co., Ltd. (Kunming, China), a voucher specimen was deposited at the innovative drug research group of Xishuangbanna Tropical Botanical Garden, Chinese Academy of Sciences (No. 20201036EW). An amount of 50 g PAH was extracted with 0.5 L of 85% ethanol under reflux three times, each time for 2 h. The obtained ethanol extracts were mixed, and the organic solvent was removed under reduced pressure to obtain a dry ethanol extract of PAH (PAHEE, 6.7 g). Then, 1.0 g PAHEE was dissolved in 30 mL purified water, and in accordance with the polarity, petroleum ether, ethyl acetate, and n-butanol were used to extract separately to obtain the petroleum ether fraction (PEF), ethyl acetate fraction (EAF), and n-butanol fraction (NBF), respectively.

### 4.3. Determination of Total Flavonoids Content

The total flavonoids content of PAHEE and its fractions were evaluated by using colorimetric method according to the literature, and rutin was considered as an equivalent [47,48]. A 20 μL sample in PBS (0.1 M, pH 6.8) and 60 μL of 5% sodium nitrite solution were mixed in test tubes, and incubated at room temperature for 6 min. Next, 60 μL 10% aluminum nitrate solution was added to the mixture to continue the reaction for 5 min. Then, 400 μL NaOH (1 M) was added to the mixture above, and incubated at room temperature for 20 min again. After the experiment, the OD value of the supernatants were measured at 510 nm, and the total flavonoids content in the samples was calculated by the standard curve constructed with rutin. The data were expressed as rutin equivalents (mg) per dry weight of fractions (g).

### 4.4. Determination of Total Phenolics Content

The total phenolic content of PAHEE and its fractions were estimated by using colorimetric method according to the literature, and rutin was considered as an equivalent [49]. A 20 μL sample in PBS (0.1 M, pH 6.8) and 500 μL Folin–Ciocalteu reagent (1 M) were mixed in test tubes, and incubated at room temperature for 4 min. An amount of 400 μL Na_2_CO_3_ (0.5 M) was then added to the mixture to continue the reaction for 60 min. After the experiment, the OD value of supernatants was measured at 760 nm in triplicate, and total phenolic content was calculated through a standard curve constructed using gallic acid. The data were expressed as gallic acid equivalents (mg) per dry weight of fractions (g).

### 4.5. α-Glucosidase Inhibition Experiments

The metabolic enzymes such as *α*-glucosidase and *α*-amylase are significant enzymes in diabetes since they are involved in food hydrolyzing activities that regulate postprandial blood glucose levels [50,51]. The *α*-glucosidase inhibition assay was carried out on the basis of Zhao et al. [52], with minor modifications. A 10 μL sample and 50 μL of *α*-glucosidase solution (0.1 u/mL) were mixed and incubated at 37 °C for 10 min, and 10 μL PBS considered as a blank control. Next, 40 μL pNPG (5 mmol/L) was added to the mixture above, and the reaction system was incubated at 37 °C for 20 min before being stopped by 50 μL Na_2_CO_3_ solution (0.1 mol/L). The absorbance of the reaction mixture was measured at 405 nm by a microplate reader (SpectraMax190, Molecular Devices, Silicon Valley, America).
Inhibition rate (%) = (Oc − Os)/Oc × 100%,(1)
where Oc is the OD value of the blank control, Os is the OD value of the tested samples, and the analysis was performed in triplicate.

### 4.6. Kinetic Analysis

The kinetic analysis of PAHEE and its fractions were carried out with final substrate concentrations of 1, 2, 3, 4, 5 µmol/L, PAHEE concentrations of 2.5, 5.0, and 10.0 μg/mL, PEF, EAF, and NBF concentrations of 1.25, 2.5, and 5.0 μg/mL, respectively, and *α*-glucosidase concentrations of 0.05, 0.10, 0.15, 0.20, 0.25 u/mL. The type of inhibition was determined by Lineweaver–Burk plot (the inverse of velocity (1/v) against the inverse of the substrate concentration (1/[S])).

### 4.7. Glucose Uptake and Cell Viability Assays

The differentiation process of 3T3-L1 adipocytes was carried out as shown in Figure 3a, and the mature adipocytes were then inoculated in a 96-well plate at 5 × 10^4^ cells per well, and the experiment was started 24 h later. The 3T3-L1 adipocytes were divided into model group (blank control), insulin group (250 ng/mL, positive control), and sample groups (20, 40, 80, and 160 μg/mL). After 24 h of administration, 10 μL medium was used to measure the glucose content. Cell viability was detected by CellTiter 96^®^ AQueous One Solution Reagent according to the manufacturer’s instructions after the glucose uptake experiment [53]. An amount of 20 µL CellTiter 96^®^ AQueous One Solution Cell Proliferation Assay reagent was added to the wells of the experiment plate, and then incubated at 37 °C for 180 min before the absorbance was measured at 490 nm, and the relative cell viability was presented after being normalized to the model group. (Samples were dissolved to 160 mg/mL by using DMSO and diluted to various concentrations (20, 40, 80, 160 μg/mL) in high glucose DMEM before the experiment.)
Cell viability (%) = Os/Oc × 100%,(2)
where Oc is the OD value of the blank control, Os is the OD value of the tested samples, and the analysis was performed in triplicate.

### 4.8. HPLC-MS Analysis of EAF

Agilant 1290uplc liquid chromatography equipped with Agilant mass spectrometry (MS) qtof6550 was used to detect and analyze the component of EAF. A Shimadzu InertSustain C18 column (100 × 2.1 mm, 2 μm) was used for HPLC (column temperature: 35 °C, flow rate: 0.3 mL/min). Scanning mode, data-independent analysis (100–1500 m/z), sheath gas temperature 500 °C, and sheath gas flow rate 12 L/min were used in mass MS conditions. The HPLC analysis was performed by step-gradient method, and the mobile phase with solvent A (acetonitrile) and solvent B (water) was as follows: 10% of eluent A at 0 min, 25% at 15 min, 40% at 30 min, and 55% at 45 min. The HPLC-MS chromatogram was preprocessed and compared with the TCM database, and the data came from the analytical results of TCM database.

### 4.9. Molecular Docking of Candidate Compounds on α-Glucosidase

In silico molecular docking was used to investigate the interactions between candidates and *α*-glucosidase [54]. The structure of halomonas *α*-glucosidase (PDB ID: 3WY1) was obtained from the Online Protein Data Bank [55], and the three-dimensional structures of the ligands were downloaded from Pubchem or MarvinSketch. Complexed ligands and water molecules in the crystal structure of *α*-glucosidase were virtually removed by pyMOL Win application (pyMOL, version: 2.2.0). AutoDock Tools (ADT, version: 1.5.6) was used to accomplish molecular docking in silico [56]. The cubic grid box dimensions of *α*-glucosidase were defined as x = 82, y = 86, and z = 126 Å with spacing of 0.681 Å. Finally, the PyMOL molecular graphics system (version 2.2.0) was used to visualize ligand–enzyme interactions.

### 4.10. DPPH Free Radical Scavenging Assay

The DPPH assay was conducted in accordance with Tsamo et al. [57]. Initially, 180 µL DPPH radical solution was mixed with 20 µL sample in PBS (0.1 M, pH 6.8), and the concentrations of PAHEE and its fractions were 12.5, 25.0, 50.0, 100.0, 200.0, 400.0, 800.0, 1600.0, and 3200.0 μg/mL, respectively. The mixtures were then kept in the dark for 30 min at room temperature. The absorbance was measured using a microplate reader at 517 nm, and ascorbic acid was used as a positive control.
Scavenging activity (%) = (Oc − Os)/Oc × 100%,(3)
where Oc is the OD value of the blank control, Os is the OD value of the tested samples. The analysis was performed in triplicate and the results were described as IC_50_ value.

### 4.11. ABTS Radical Cation Scavenging Assay

The ABTS radical cation scavenging assay was conducted in accordance with instructions for ABTS kit (Suzhou Comin Biotechnology Co., Ltd.; Suzhou, China). Initially, a 10 µL sample (800 μg/mL) and 190 µL of ABTS working reagent were mixed, and incubated at room temperature for 5 min. The absorbance was then measured by using a microplate reader at 734 nm.
Scavenging activity (mmol Trolox/L) = (Oc − Os + 0.0012)/0.7021,(4)
where Oc is the OD value of the blank control, Os is the OD value of the tested samples, and the analysis was performed in triplicate.

### 4.12. Ferric Reducing Antioxidation Power

The Ferric reducing antioxidation power (FRAP) assay radical scavenging was conducted in accordance with instructions for FRAP kit (Suzhou Comin Biotechnology Co., Ltd.; Suzhou, China). Initially, a 10 µL sample (800 μg/mL) and 190 µL of FRAP working reagent were mixed, and incubated at room temperature for 20 min. The absorbance was then measured by using a microplate reader at 593 nm.
Scavenging activity (mmol Trolox/L) = (Os − Oc − 0.0134)/0.1246,(5)
where Oc is the OD value of the blank control, Os is the OD value of the tested samples, and the analysis was performed in triplicate.

### 4.13. Statistical Analysis

IBM SPSS Statistics for Windows, version 21.0 (IBM Corp., Armonk, NY, USA) was used to analyze all of the data, and the results were expressed as the average of the three measurements ± SD. Multigroup comparisons of the means were carried out by one-way analysis of variance test with post hoc contrasts by Student–Newman–Keuls test. Differences were considered significant when * *p* < 0.05, ** *p* < 0.01, *** *p* < 0.001.

## 5. Conclusions

Our study indicated that PAH contain rich sources of natural hypoglycemic molecules and antioxidants, such as flavonoids and phenolics. The results of total flavonoids content and total phenolics content were EAF > PEF > NBF > PAHEE, and the results of *α*-glucosidase inhibitory activity was EAF > PEF > PAHEE > NBF. In addition, glucose uptake experiment showed that NBF possessed significant promotion ability on glucose uptake rate of 3T3-L1 adipocytes. HPLC-MS analysis and molecular docking results proved the interactions between candidates and *α*-glucosidase. (Flavonoids: 3,5,7,3′,4′,-pentahydroxyflavone-3-L-rhamnoside, cyanidin, agathisflavone, delphinidin 3-*O*-*β*-D-galactoside, avicularin, delphinidin-3-arabinoside, etc.) Consistent with the results of *α*-glucosidase inhibitory activity, the antioxidation capacities of PAHEE and its fractions on DPPH, ABTS, and FRAP were EAF > PEF > PAHEE > NBF. In general, the best antioxidation capacity of EAF might be related to its abundant contents of flavonoids and phenolics, and the hypoglycemic activity of PAH might be related to the inhibition of α-glucosidase activity (EAF, PEF, and NBF) and the promotion of glucose uptake in 3T3-L1 adipocytes (NBF). Our research indicated that EAF and/or NBF might be used as potential natural hypoglycemic agent and/or antioxidant for further new drug research and development.

## Figures and Tables

**Figure 1 molecules-27-03381-f001:**
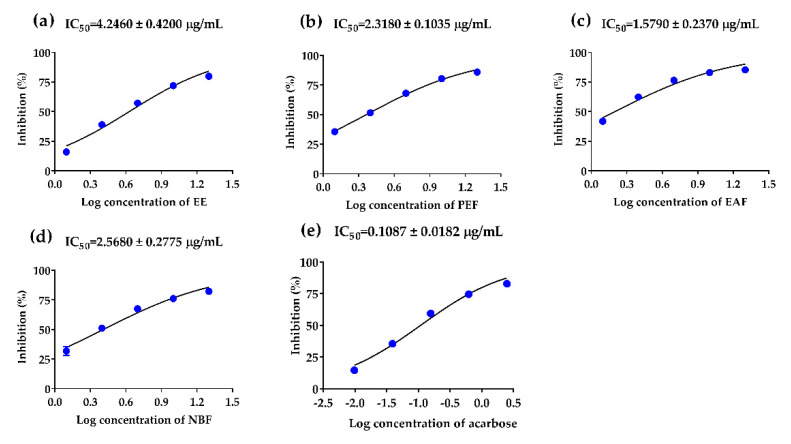
α-Glucosidase inhibitory effects of PAHEE and its fractions. (**a**) Log concentration–inhibition rate fitting curve of PAHEE. (**b**) Log concentration–inhibition rate fitting curve of petroleum ether fraction (PEF). (**c**) Log concentration–inhibition rate fitting curve of ethyl acetate fraction (EAF). (**d**) Log concentration–inhibition rate fitting curve of n-butanol fraction (NBF). (**e**) Log concentration–inhibition rate fitting curve of acarbose. Calculated the IC_50_ value of different groups by Statistical Product and Service Solutions (SPSS, version: 21.0, International Business Machines Corporation, New York, USA), and all values are mean ± SD from a least three independent experiments.

**Figure 2 molecules-27-03381-f002:**
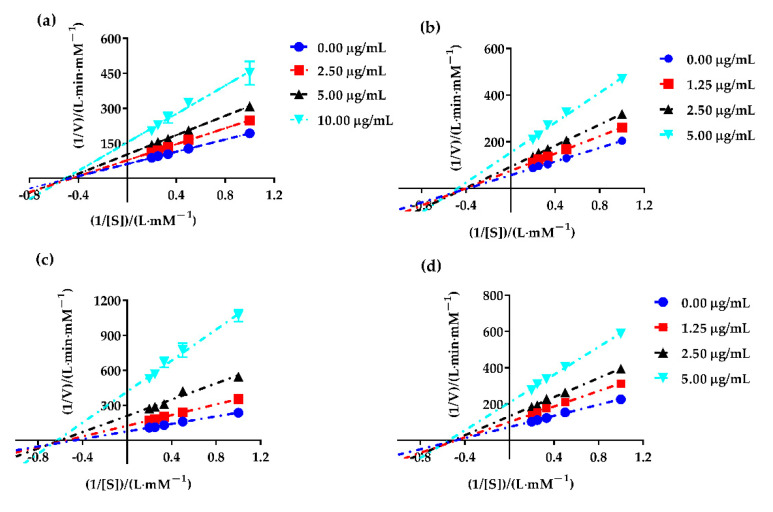
The Lineweaver–Burk plot of PAHEE and its fractions. (**a**) 1/[S]–1/V fitting curve of PAHEE. (**b**) 1/[S]–1/V fitting curve of PEF. (**c**) 1/[S]–1/V fitting curve of EAF. (**d**) 1/[S]–1/V fitting curve of NBF, and all values are mean ± SD from a least three independent experiments.

**Figure 3 molecules-27-03381-f003:**
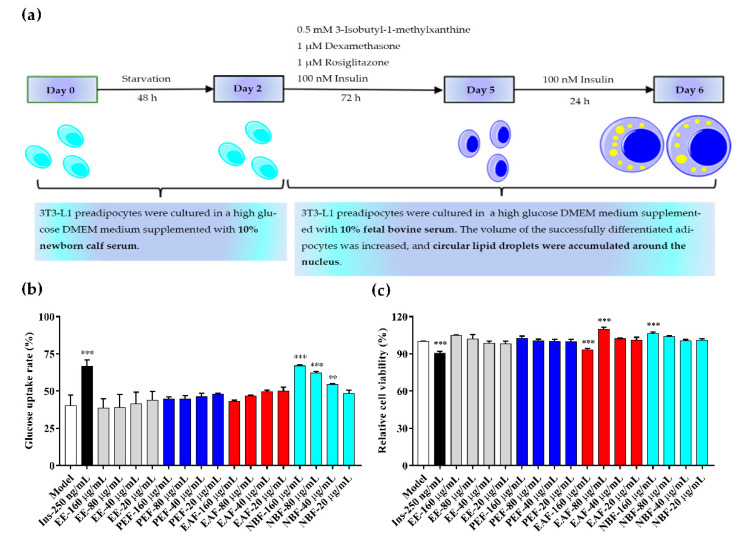
Adipocyte differentiation process and glucose uptake, cell viability test results of PAHEE and its fractions. (**a**) Schematic diagram of 3T3-L1 adipocytes differentiation. (**b**) Glucose uptake rate of different groups (Model, Ins, PAHEE, PEF, EAF, NBF), insulin (Ins) was used as a positive control. (**c**) Cell viability of different groups (Model, Ins, PAHEE, PEF, EAF, NBF). All values are mean ± SD from a least three independent experiments, and each group is compared with model group, Significant are denoted by symbols: ** *p* < 0.01, and *** *p* < 0.001.

**Figure 4 molecules-27-03381-f004:**
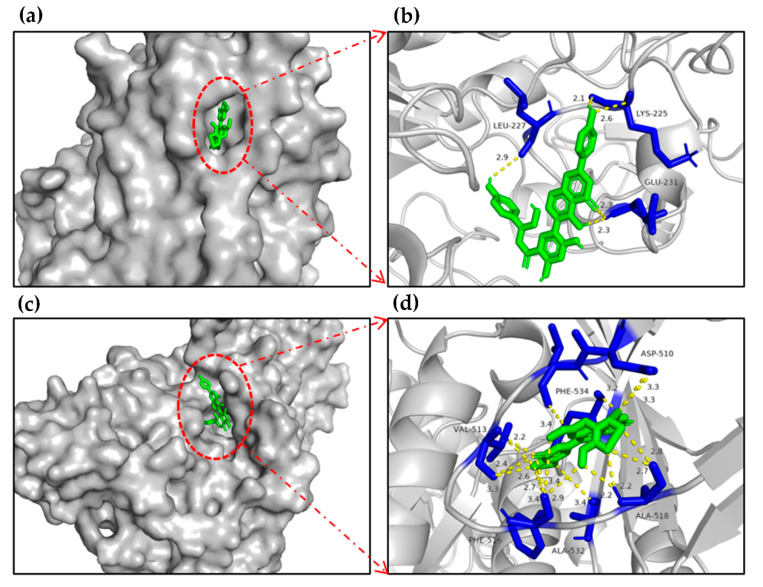
Molecular docking schemes of agathisflavone and delphinidin 3-*O*-*β*-D-galactoside on *α*-glucosidase of 3WY1. (**a**) The surface structure of 3WY1-agathisflavone. (**b**) The binding site structure of 3WY1-agathisflavone. (**c**) The surface structure of 3WY1-delphinidin 3-*O*-*β*-D-galactoside. (**d**) The binding site structure of 3WY1-delphinidin 3-*O*-*β*-D-galactoside.

**Figure 5 molecules-27-03381-f005:**
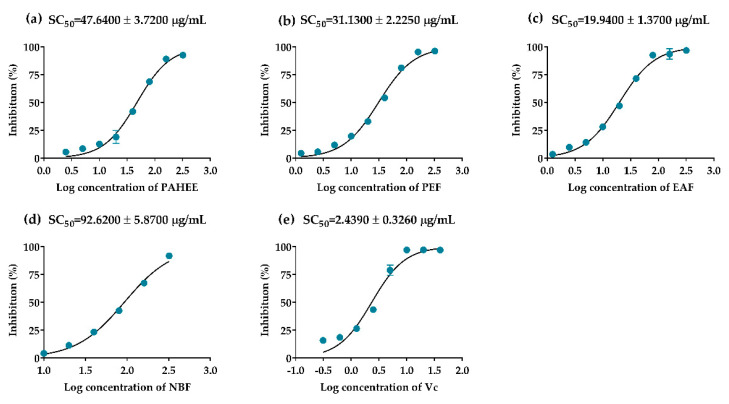
The scavenging effects of PAHEE and its fractions on DPPH-free radical. (**a**) Log concentration–scavenging rate fitting curve of PAHEE. (**b**) Log concentration–scavenging rate fitting curve of PEF. (**c**) Log concentration–scavenging rate fitting curve of EAF. (**d**) Log concentration–scavenging rate fitting curve of NBF. (**e**) Log concentration–scavenging rate fitting curve of ascorbic acid. Calculated the SC_50_ value of different groups by Statistical Product and Service Solutions (SPSS, version: 21.0, International Business Machines Corporation, New York, USA), and all values are mean ± SD from a least three independent experiments.

**Table 1 molecules-27-03381-t001:** Total flavonoids content (TFC) and total phenolics content (TPC) of PAHEE and its fractions.

Samples	TFC (mg/g)	TPC (mg/g)
PAHEE	68.7 ± 0.3 ^C^	43.2 ± 0.6 ^C^
PEF	88.5 ± 4.4 ^B^	62.2 ± 1.1 ^B^
EAF	159.7 ± 2.5 ^A^	107.6 ± 2.0 ^A^
NBF	34.7 ± 2.0 ^D^	29.7 ± 0.4 ^D^

The values represent mean ± SD, *n* = 3. Polygoni Avicularis Herba ethanolic extract (PAHEE), petroleum ether fraction (PEF), ethyl acetate fraction (EAF), n-butanol fraction (NBF). Within the same column, values with different superscript capital letters are statistically different (*p* < 0.05).

**Table 2 molecules-27-03381-t002:** HPLC/MS analysis and molecular docking results of compounds detected from EAF.

Category	Compound Name	Formula	Mass (g/mol)	Reference Mass (g/mol)	RT (min)	Area (μV·s)	Score (%)	Binding Energy (kcal/mol)	References
Flavonoids	3,5,7,2′,6′-Pentahydroxyflavone	C_15_H_10_O_7_	302.0	302.0	22.7	100671889	99.4	−9.55	[25] ^a^
	5,7,2′,3′-Tetrahydroxyflavone	C_15_H_10_O_6_	286.0	286.1	27.0	65396294	96.3	−7.15	*
	3′-Methoxydaidzein	C_16_H_12_O_5_	284.1	284.1	48.2	43163766	97.0	−8.80	*
	Agathisflavone	C_30_H_18_O_10_	538.1	538.1	42.0	39711957	99.5	−11.35	[26] ^a,b^
	6-Hydroxykaempferol-7-*O*-glucoside	C_21_H_20_O_12_	464.1	464.1	17.9	67100153	98.9	−8.08	*
	Avicularin	C_20_H_18_O_11_	434.1	434.1	22.7	57090941	99.0	−10.02	[27] ^a^; [28] ^b^
	Desmanthin 2	C_28_H_24_O_16_	616.1	616.1	16.5	29675775	98.1	−8.45	*
	3,5,7,3′,4′ -Pentahydroxyflavone-3-L-rhamnoside	C_21_H_20_O_11_	448.1	448.1	24.1	26798354	96.8	−10.20	*
	3,7,4′,5-Tetrahydroxyflavone-3-L-rhamnoside	C_21_H_20_O_10_	432.1	432.1	28.5	15851984	98.8	−8.16	*
	Melicitrin	C_20_H_18_O_12_	450.1	450.1	17.6	11126608	98.9	−6.79	*
	2′′-*O*-Galloylisoorientin	C_28_H_24_O_15_	600.1	600.1	30.5	10835996	97.8	−9.78	*
	Delphinidin	C_15_H_11_O_7_	303.1	303.1	28.7	47200508	99.5	−7.50	[29] ^a,b^; [30] ^b^
	Cyanidin	C_15_H_11_O_6_	287.1	287.1	34.4	32042323	99.2	−8.03	[31] ^b^; [32] ^b^
	Pelargonidin	C_15_H_11_O_5_	271.1	271.1	35.5	27253529	97.6	−8.43	[33] ^b^
	Peonidin	C_16_H_13_O_6_	301.1	301.1	36.7	14049096	99.3	−6.65	*
	Delphinidin-3-*O*-*β*-D-galactoside	C_21_H_21_O_12_	465.1	465.1	17.9	85247661	98.9	−11.58	*
	Delphinidin-3-arabinoside	C_20_H_19_O_11_	435.1	435.1	22.7	70570638	99.0	−8.52	[34] ^b^
	Pelargonidin-3-galactoside	C_21_H_21_O_10_	433.1	433.1	28.5	19201678	99.2	/	[35] ^b^
	Delphinidin-3′-*O*-(2″-*O*-galloyl-*β*-galactoside)	C_28_H_25_O_16_	617.1	617.1	17.3	13797523	98.2	−6.29	--
Phenolics	2-Methoxy-4-(3-methoxy-1-propenyl)-phenol	C_11_H_14_O_3_	194.1	194.1	11.73	16988535	97.9	−5.14	*
Alkaloids	Leonurine	C_14_H_21_N_3_O_5_	311.2	311.2	26.7	27644760	98.6	/	[36] ^b^
	Gnoscopine	C_24_H_27_NO_6_	425.2	425.2	46.5	24023457	97.1	/	--
	8-Acetyldolaconine	C_26_H_39_NO_6_	461.3	461.3	31.8	18923730	98.6	/	--
Quinones	1,8-Dihydroxy-4-hydroxymethyl anthraquinone	C_15_H_10_O_5_	270.1	270.1	35.5	23901981	97.6	/	--
	1,6-Dihydroxy-2,4-dimethoxyanthraquinone V	C_16_H_12_O_6_	300.1	300.1	36.7	12128297	99.3	/	--
	Abieta-8,12-dien-11,14-dione	C_20_H_28_O_2_	300.2	300.2	37.8	11252041	98.6	/	--
Terpenoids	(12R)-12-Hydroxy cascarill one	C_20_H_30_O_3_	318.2	318.2	43.2	12287901	96.6	/	--

Where “^a^” represent antioxidation activity, “^b^” represent hypoglycemic activity, “*” represent reported for the first time, “/” represent had not been detected.

**Table 3 molecules-27-03381-t003:** Antioxidant activities of PAHEE and its fractions via assays for DPPH, ABTS, and FRAP.

Samples	DPPH (SC_50_/μg/mL)	ABTS (mmol Trolox/L)	FRAP (mmol Trolox/L)
PAHEE	47.64 ± 3.72 ^B^	0.83 ± 0.03 ^C^	0.35 ± 0.00 ^C^
PEF	31.13 ± 2.23 ^C^	1.12 ± 0.01 ^B^	0.54 ± 0.00 ^B^
EAF	19.94 ± 1.37 ^D^	1.30 ± 0.00 ^A^	0.76 ± 0.04 ^A^
NBF	96.62 ± 5.87 ^A^	0.42 ± 0.12 ^D^	0.14 ± 0.00 ^D^
Ascorbic acid	2.44 ± 0.33 ^E^	--	--

The values represent mean ± SD, *n* = 3. 2, 2-diphenyl-2-picrylhydrazyl (DPPH), 2, 2-azinobis-3-ethylbenzothia-zoline-6-sulfonic acid (ABTS), ferric reducing-antioxidant power (FRAP) experiment. A-E (DPPH), A-D (ABTS and FRAP) within the same column, values with different superscript capital letters are statistically different (*p* < 0.05).

## Data Availability

The data presented in this study are available on request from the corresponding author.

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
