# Peer review of "Hypoglycemic and Antioxidant Properties of Extracts and Fractions from Polygoni Avicularis Herba"

_molecules, 2022, doi:10.3390/molecules27113381_

Round 1
Reviewer 1 Report
Dear Authors
I did my detailed comments in the previous submission.
In this version, you did the recommended suggestions and the article, for me, is ready to be published.
I have nothing more to comment on.
Best regards
Author Response
Dear reviewer:
On behalf of my co-authors, we appreciate very much for your positive and constructive comments and suggestions on our manuscript entitled “Hypoglycemic and Antioxidant Properties of Fractions from Polygoni Avicularis Herba Ethanolic Extract”. (Manuscript ID: 1590575; Resubmission manuscript ID: 1684032).
Comments and Suggestions for Authors
Dear Authors
I did my detailed comments in the previous submission.
In this version, you did the recommended suggestions and the article, for me, is ready to be published.
I have nothing more to comment on.
Reply: Thank you for your kind help with our manuscript.
Reviewer 2 Report
The manuscript entitled "Hypoglycemic and Antioxidant Properties of Extract and Fractions from Polygoni Avicularis Herba"
describes research focusing on hypoglycemic and antioxidant activity of Polygonum aviculare L., herba. and is improved version of previously
submitted manuscript.
The research included determination of antioxidant activity using ABTS, DPPH, FRAP tests,
determination of total flavonoids, inhibition of alpha glucosidase, alpha-amylase and promotion of glucose uptake in 3T3-L1 adipocytes.
Some of the fractions were found to be active as glucosidase and amylase inhibitors and promoted glucose uptake in 3T3-L1 adipocytes,
particularly n-butanol fraction.
The extract was also investigated by LC-MS to identify the costituents and molecular docking was performed to pick
which compound may be responsible for the alpha glucosidase inhibition. The molecular docking results
showed that agathisflavone and delphinidin 3-O-β-D-galactoside possessed the lowest binding energy
with α-glucosidase. The paper provides interesting results, nevertheless the activity of the selected compouds was not verified in vitro.
Therefore authors should obligatorily discuss well the abvailable literature on activity of these compounds:
E.g.: https://ift.onlinelibrary.wiley.com/doi/10.1111/1750-3841.15438, https://www.researchgate.net/publication/347054732_STRUCTURE-ACTIVITY-RELATIONSHIP_OF_THE_POLYPHENOLS_INHIBITION_OF_a-AMYLASE_AND_a-GLUCOSIDASE
and other.
"polyphenolics" throughout text should be changed to phenolics. The method based on Folin-Ciocalteu reagents,
allows to determine all phenols, not only polyphenols.
e.g. in Table 3 "Other polyphenolics 2-Methoxy-4-(3-methoxy-1-propenyl)-phenol" is innapropriate, there is just one phenol group.
Author Response
Dear reviewer:
On behalf of my co-authors, we appreciate very much for your positive and constructive comments and suggestions on our manuscript entitled “Hypoglycemic and Antioxidant Properties of Extract and Fractions from Polygoni Avicularis Herba”. (Resubmission manuscript ID: 1684032).
We have carefully-considered the comments from you, which help us to improve the manuscript substantially. We have revised the content of the manuscript according to your valuable suggestions. The changes in the revised manuscript are highlighted in red. We also provide a point-to-point response to the comments with references to the changes made in the text, as follows:
Comments and Suggestions for Authors
The manuscript entitled "Hypoglycemic and Antioxidant Properties of Extract and Fractions from Polygoni Avicularis Herba" describes research focusing on hypoglycemic and antioxidant activity of Polygonum aviculare L., herba. and is improved version of previously submitted manuscript.
The research included determination of antioxidant activity using ABTS, DPPH, FRAP tests, determination of total flavonoids, inhibition of alpha-glucosidase, alpha-amylase and promotion of glucose uptake in 3T3-L1 adipocytes. Some of the fractions were found to be active as glucosidase and amylase inhibitors and promoted glucose uptake in 3T3-L1 adipocytes, particularly n-butanol fraction. The extract was also investigated by LC-MS to identify the costituents and molecular docking was performed to pick which compound may be responsible for the alpha glucosidase inhibition. The molecular docking results showed that agathisflavone and delphinidin 3-O-β-D-galactoside possessed the lowest binding energy with α-glucosidase.
- Comment: The paper provides interesting results, nevertheless the activity of the selected compounds was not verified in vitro. Therefore authors should obligatorily discuss well the available literature on activity of these compounds:
E.g.:https://ift.onlinelibrary.wiley.com/doi/10.1111/1750-3841.15438, https://www.researchgate.net/publication/347054732_STRUCTURE-ACTIVITY RELATIONSHIP_OF_THE_POLYPHENOLS_INHIBITION_OF_a-AMYLASE_AND_a-GLUCOSIDASE
- Reply: According to your valuable suggestions, we supplemented the existing reports of these compounds on hypoglycemic activity in the discussion section of the manuscript, as follows: “According to literatures, some of flavonoids detected by HPLC-MS in our study have been reported to have potential hypoglycemic and antioxidant activity. On the one hand, the results showed that agathisflavone, avicularin, cyanidin, and pelargonidin exhibited significantly digestive enzymes inhibitory activity (α-glucosidase and / or α-amylase), and agathisflavone, avicularin inhibited α-glucosidase with IC50 values of 11.4 ± 0.9 µmol/L, 69.8 mg/L, respectively. In addition, delphinidin-3-arabinoside have the potential to modulate dipeptidyl peptidase-IV and its substrate GLP-1, to increase insulin secretion [28-36]. On the other hand, previous research suggested that 3,5,7,2',6'-pentahydroxyflavone with the hydroxyl group in the B ring possessed a potent activity against lipid peroxidation [27]. In summary, the results suggested that PAH might possess great hypoglycemic and antioxidant activities, and its hypoglycemic activity might be related to the inhibition of digestive enzymes, and the promotion of insulin secretion.”
- Comment: Other "polyphenolics" throughout text should be changed to phenolics. The method based on Folin-Ciocalteu reagents, allows to determine all phenols, not only polyphenols. e.g. in Table 3 "Other polyphenolics 2-Methoxy-4-(3-methoxy-1-propenyl)-phenol" is inappropriate, there is just one phenol group.
- Reply: Thank you for your kind reminding, according to your valuable suggestion, we revised the “polyphenolics” in the manuscript into “phenolics”.
Reviewer 3 Report
In my opinion this paper needs several corrections:
2 – It should be „… Extracts and its Fractions …”.
14 – Throughout entire paper, it should be „ABTS assay”.
16 – It should be “ the highest content of total flavonoids and total phenolics …”.
16, 57, 81, Table 1 – The results should be reported with one digital after decimal point.
16/17 – It should be “… mg galic acid equivalents/g and …. mg rutin equivalents/g.
36 – It should be “The main bioactive compounds …, and quinones [13]”.
39 – Arbutus is a plant nor flavonoid. Olivine is a mineral. Astragaloside is a triterpene. “albizin 1”? Methyl gallate is phenolic acid derivative.
62 – It should be “… determined using FRAP and ABTS assays.”
83 – It should be “exhibited” not “had”.
93 – TPC means “total phenolic content”.
98/100, 224/227 – Not with italic.
135 – What does it mean “of different groups”?
168 – It should be “that flavonoids agathisflavone and delphinidin- …. “.
189 – “is widely associated”?
289 – “p-“ with italic.
309 – “standard material”?
315/316 – It should be rephrased.
386 – It should be “DPPH radical solution”.
Author Response
Dear reviewer:
On behalf of my co-authors, we appreciate very much for your positive and constructive comments and suggestions on our manuscript entitled “Hypoglycemic and Antioxidant Properties of Extract and Fractions from Polygoni Avicularis Herba”. (Resubmission manuscript ID: 1684032).
We have carefully-considered the comments from you, which help us to improve the manuscript substantially. We have revised the content of the manuscript according to your valuable suggestions. The changes in the revised manuscript are highlighted in red. We also provide a point-to-point response to the comments with references to the changes made in the text, as follows:
Comments and Suggestions for Authors
In my opinion this paper needs several corrections:
- Comment: 2 – It should be „… Extracts and its Fractions …”.
- Reply: According to your valuable suggestion, we revised “Extract” into “Extracts”.
- Comment: 14 – Throughout entire paper, it should be “ABTS assay”.
- Reply: According to your valuable suggestion, we revised “ABTS•+” into “ABTS”.
- Comment: 16 – It should be “the highest content of total flavonoids and total phenolics …”.
- Reply: According to your valuable suggestion, we have revised “polyphenolics” into “phenolics”.
- Comment: 16, 57, 81, Table 1 – The results should be reported with one digital after decimal point.
- Reply: Thank you for your valuable suggestions, we have revised the results in rows 16, 57, 81, Table 1 with one digital after decimal point.
- Comment: 16/17 – It should be “… mg galic acid equivalents/g and …. mg rutin equivalents/g.
- Reply: According to your valuable suggestion, we have revised “… values of 159.74 ± 2.53 and 107.55 ± 1.96 16 mg/g, respectively, rutin and gallic acid were considered as the equivalents of flavonoids and polyphenolics respectively.” into “… values of 159.7 ± 2.5 mg rutin equivalents/g and 107.6 ± 2.0 mg galic acid equivalents/g, respectively.”.
- Comment: 36 – It should be “The main bioactive compounds …, and quinones [13]”.
- Reply: According to your valuable suggestion, we have revised “The main chemical components obtained in PAH were flavonoids, phenolic acids, alkaloids, terpenes, sterols, quinones, other phenylpropanoids, and so on [13].” into “The main bioactive components obtained in PAH were flavonoids, phenolic acids, alkaloids, terpenes, sterols, quinones [13].”.
- Comment: 39 –Arbutus is a plant nor flavonoid. Olivine is a mineral. Astragaloside is a triterpene. “albizin 1” ? methyl gallate is phenolic acid derivative.
- Reply: Thank you for your kind reminding, we removed the inappropriate compounds in this manuscript.
- Comment: 62 – It should be “… determined using FRAP and ABTS assays.”
- Reply: “… by ferric reducing-antioxidant power (FRAP) experiment, and 2, 2-azinobis-3-ethylbenzothia-zoline-6-sulfonic acid (ABTS•+).” into “determined by FRAP and ABTS assays”.
- Comment: 83 – It should be “exhibited” not “had”.
- Reply: According to your valuable suggestion, we have revised “had” into “exhibited”.
- Comment: 93 – TPC means “total phenolic content”.
- Reply: According to your valuable suggestion, we have revised “total polyphenolics content” into “total phenolics content”.
- Comment: 98/100, 224/227 – Not with italic.
- Reply: According to your suggestion, we corrected the error in the manuscript.
- Comment: 135 – What does it mean “of different groups”?
- Reply: Thank you for your kind reminding, we added instructions about “of different groups”, as follows “of different groups (Model group, positive control group, PAHEE group, PEF group, EAF group, and NBF group).”.
- Comment: 168 – It should be “that flavonoids agathisflavone and delphinidin- …. “.
- Reply: According to your valuable suggestion, we have revised “… that agathisflavone (flavonoids) and delphinidin 3-O-β-D-galactoside (flavonoids)…” into “… that flavonoids agathisflavone and delphinidin 3-O-β-D-galactoside…”.
- Comment: 189 – “is widely associated”?
- Reply: According to your valuable suggestion, we revised “… is widely associated to the antioxidant capacity of natural products” into “is widely used to determine the antioxidant capacity of natural products”.
- Comment: 289 – “p-”with italic.
- Reply: Thank you for your kind reminding, we revised the error in the manuscript.
- Comment: 309 – “standard material”?
- Reply: Thank you for your kind reminding, we revised “standard material” into “an equivalent”.
- Comment: 315/316 – It should be rephrased.
- Reply: According to your valuable suggestion, we revised “… the OD value of supernatants were measured at 510 nm in triplicate, and total flavonoids content was determined through a standard curve constructed by rutin.” into “…the OD value of the supernatants were measured at 510 nm, and the total flavonoids content in the samples was calculated by the standard curve constructed with rutin.”.
- Comment: 386 – It should be “DPPH radical solution”.
- Reply: According to your valuable suggestion, we revised “DPPH solution” into “DPPH radical solution”.
Round 2
Reviewer 2 Report
The manuscript was improved according to the remarks
Author Response
Comments and Suggestions for Authors
The manuscript was improved according to the remarks.
Response 1: Thank you for your kind help with our manuscript.
We carefully rechecked the English language and style of this manuscript, and made some minor revisions, as follows:
Point 1: Row 18, we revised “the best” into “a potent”.
Point 2: Row 54, we revised “The previous” into “Previous”.
Point 3: Row 62, we revised “evidence” into “evidences”.
Reviewer 3 Report
The authors corrected this paper properly taken under considerations all my comments. Therefore, I can accept it now.
Author Response
Comments and Suggestions for Authors
The authors corrected this paper properly taken under considerations all my comments. Therefore, I can accept it now.
Response 1: Thank you for your kind help with our manuscript.
This manuscript is a resubmission of an earlier submission. The following is a list of the peer review reports and author responses from that submission.
Round 1
Reviewer 1 Report
COMMENTS
General
- a graphical abstract can contributes to the article interface
Abstract
- the Abstract is too long
Introduction
- rows 39-45, too long phrases, there are need reformulation
- row 56 Alzheimer with A not a
- row 57, organ fibrosis, please be more exactly (liver fibrosis, etc)
Results
- the values contain too many decimals while the methods are chemometrically established with 2 max 3 decimals but not 4
- row 83 and others, all significant changes mentioned and described with experimental data need P values
- dot after the word Figure eg Figure. 1. is not usually; Figure 1. is more frequently used;
- all tables and figures need a consistent revision in line of titles and captions; the readers must understand the values without other texts; the figures and captions must be self-explanatory;
- row 108 eg. The inhibitory effect of (…) on what? more explanations are needed;
- Figure 2 - what are the differences between plots?
between P and <, tab space
Discussion
- this chapter is too short, a comparison with other effects of similar molecules, a short literature review about ultrastructural changes induced by PAH or other species of Polygonum genre will contribute to the high importance of the results;
Reviewer 2 Report
The manuscript "Hypoglycemic and Antioxidant Properties of Fractions from Polygoni Avicularis Herba Ethanolic Extract" shows relevant data associated with Polygonum aviculare L. extracts which are used in traditional Chinese medicine. This medicinal plant is, also, validated by the European Medicines Agency.
All my recommendations to the authors and questions were made in the pdf that will be attached to this review.
In general, the work has merit and should be published.
.

Reviewer 3 Report
Actually it is good written article and it can be published in the Molecules.
Minor concerns:
Table 1 - I think it is better to devide it and place the results in proper sections (extract content and antioxidative action). In this table there is no statistical comparision between groups. If ascorbic acid didn't work in ABTS and FRAP why another positive control was not used? And it is better to explain abbriviations again in all tables for better perception.
In Glucose uptake and cell viability assays it is not written how samples were dilluted prior to introduction into wells.
Reviewer 4 Report
The manuscript entitled "Hypoglycemic and Antioxidant Properties of Fractions from Polygoni Avicularis Herba Ethanolic Extract"
describes research focusing on hypoglycemic and antioxidant activity of Polygonum aviculare L., herba.
The research included determination of antioxidant activity using ABTS, DPPH, FRAP tests, determonation of total flavonoids, inhibition of alpha glucosidase,
alpha-amylase and promotion of glucose uptake in 3T3-L1 adipocytes. SOme of the fractions were found to be active as glucosidase and amylase inhibitors
and promoted glucose uptake in 3T3-L1 adipocytes, particularly n-butanol fraction.
The extract was also investigated by LC-MS to identify the costituents and molecular docking was performed to pick
which compound may be responsible for the alpha glucosidase inhibition. The molecular docking results
showed that agathisflavone and delphinidin 3-O-β-D-galactoside possessed the lowest binding energy
with α-glucosidase. Nevertheless the activity of the selected compouds was not verified in vitro.
The paper provides interesting results however it seeme a bit incomplete, since the activity of the picked
compounds thet could be mainly responsible for the activity was not verified in vitro.
By the way, why molecular docking to alpha amylase was not performed?
Some other comments:
- "poygoni avicularis herba" - italic
- abstract, line 18: please provide in equivalents of what, the flavonoids and phenol content is expressed
- abstract & line 84 - please explain abbreviation
- Table 1 - please explain abbreviation in table footer
- line 161 - why activity of those compounds was not tested in vitro?
- Figure 4 - the figure should be placed next to text on molecular docking
- line 260 -266 - the data was not verified in vitro!
Reviewer 5 Report
Due to low scientific quality this paper cannot be accepted.
Language is not acceptable. This paper must be corrected by a native speaker in English who is a chemist.
16 – “assays were used for test”?
16 and entire paper – It should be “phenolics” or “pnenolic compounds” instead of “phenols”.
16 – Flavonoids belong to flavonoids.
18 – It should be “mg/g”.
18 – The results should be reported with one digital after decimal point.
24, 148 – Cyanidins belong to flavonoids!
- 26 and entire document – Radical is scavenged not inhibited. The authors should use a term of “SC50”.
26 – Results with two digitals after decimal point.
26, Tble 1 – The results of ABTS and FRAP assays must be reported as mmol Trolox per mass unit.
29 – Too trivial sentence.
30 – The abbreviations need to be explained.
40/41, 144 – Phenolic acids and flanonoids belong to phenylopropanoids.
43/44 – Gallic acid and caffeic acid phenolic acids not flavonoids.
52 – Flavonoids belong to phenolic compounds.
56 – “alzheimer”?
62 – What type of assays?
66/75 – “experiments”?
84/85, 183 – Too many digitals after decimal point.
95, 101/102 – New title is needed.
135 – “insulin group”?
192 and other lines – It should be “ABTS radical cation”.
288 – “purified water”?
293 – It should be “using colorimetric method”.
301 “represented”?
310 – “determined”?
332 – “The absorbance was detected”?!
348 – An HPLC system must be described.
355 – It should be “mass spectrometry”.
371 – The concentration of the extract or its fraction?
379 – It should be “ABTS•+”.
396 – The used post-hoc test must be mentioned.
Conclusions is like as Summary.
Table 2 – The elucidation of chemical structure of the compounds that were not before reported must be completed.